# Pathways of Mycotoxin Occurrence in Meat Products: A Review

Jelka Pleadin [1] , Tina Lešić [1] , Dragan Milićević [2] , Ksenija Markov [3] , Bojan Šarkanj [4] , Nada Vahčić [3] , Ivana Kmetič [3] and Manuela Zadravec [5,*]

1 Laboratory for Analytical Chemistry, Croatian Veterinary Institute, Savska Cesta 143, 10000 Zagreb, Croatia; pleadin@veinst.hr (J.P.); lesic@veinst.hr (T.L.)
2 Institute of Meat Hygiene and Technology, Kaćanskog 13, 11040 Belgrade, Serbia; dragan.milicevic@inmes.rs
3 Faculty of Food Technology and Biotechnology, University of Zagreb, Pierottijeva 6, 10000 Zagreb, Croatia; kmarko@pbf.hr (K.M.); nvahcic@pbf.hr (N.V.); ikmetic@pbf.hr (I.K.)
4 Department of Food Technology, University North, Dr. Žarko Dolinar Square 1, 43000 Koprivnica, Croatia; bsarkanj@unin.hr
5 Laboratory for Feed Microbiology, Croatian Veterinary Institute, Savska Cesta 143, 10000 Zagreb, Croatia
* Correspondence: zadravec@veinst.hr

**Abstract:** Documented cases of mycotoxin occurrence in meat products call for further research into potential contamination sources, especially given an ever more increasing consumption of these nutritionally rich products. These foodstuffs can be contaminated with mycotoxins through three pathways: contaminated spices and other raw materials, mycotoxin-producing moulds present on the surface of dry-cured meat products, and carry-over effect from farm animals exposed to contaminated feed. In order to establish meat products' mycotoxin contamination more precisely, the concentrations of all mycotoxins of relevance for these products should be determined. This manuscript reviews data on major mycotoxins present in different types of meat products, and discusses the contamination pathways, contamination levels and control & preventative measures.

**Keywords:** mycotoxins; meat products; contaminated spices; surface moulds; carry-over effect; control and preventative measures

## 1. Introduction

Mycotoxins are a heterogeneous group of toxic substances with diverse and strong pharmacological and toxic effects on humans and animals [1]. They represent secondary fungal metabolites produced during mould growth and contaminate a variety of foodstuffs. Foodstuffs recognized as the riskiest are grains, rice, beans, coffee, wine, fruits, nuts, spices, eggs, and meat products. The problem is that their occurrence is not fully preventable in spite of research efforts and mitigation strategies.

Harmful effects observed in humans and animals include carcinogenicity, teratogenicity, immune toxicity, neurotoxicity, hepatotoxicity, nephrotoxicity, reproductive and developmental toxicity, indigestion and so forth [2]. Mycotoxins, together with viral and bacterial agents, can still compromise food safety and hygiene [3,4]. While data released by the FAO in 1999 show a 25% mycotoxin contamination of cereals, more recent data show that contamination to be significantly higher (about 60–80%) [5]. Studies show that global mycotoxin prevalence in crops varies depending on many factors, such as the type of mycotoxin and the analytical or reporting methods used. Additionally, the extensive 2008–2017 research in nearly a hundred countries, done on different cereals and their products, proved a strong association between mycotoxin occurrence and climatic conditions. Mycotoxins were detected in the vast majority of samples (88% positive on at least one mycotoxin) and were frequently co-occurring (64% of samples positive on at least two mycotoxins) [6].

As for the contamination of meat products, it was evidenced that these foodstuffs can be contaminated through three pathways: (i) contaminated spices and other raw

materials; (ii) mycotoxin-producing moulds present on the surface of dry-cured meat products; and (iii) the carry-over effect from farm animals exposed to contaminated feed [7–13]. Mycotoxins tend to persist in raw materials and final products and accumulate in the human body, causing severe health impairments coming as a result of contaminated food consumption [14–16]. In order to prevent the contamination, advanced manufacturing of safe and high-quality meat products should include the prevention and control of all critical points [7], above all that of feed intended for food animals [17] and that of final products.

Traditional meat products, such as fermented sausages, liver and blood pudding sausages, dry rack & blade, bacon, pancetta, ham, etc., are consumed worldwide, and consumers are therefore increasingly demanding higher-level quality and safety of these products. Documented cases of mycotoxin occurrence in different types of meat products requires further research into potential contamination sources, especially given an ever more increasing consumption of these nutritionally rich products. Differences in meat products' quality and safety seen during the production are not only to be attributed to different household recipes, but also to greatly varying hygienic and environmental settings conditioning the growth of specific micro-flora [7,12,18,19].

This manuscript reviews data on major mycotoxins present in meat products, and discusses the contamination pathways, contamination levels in different types of meat products and control and preventative measures.

## 2. Major TMP Mycotoxins

The most toxic mycotoxins commonly found in meat are aflatoxin $B_1$ ($AFB_1$) and ochratoxin A (OTA). $AFB_1$ is the most potent mammalian liver carcinogen, and is classified by the International Agency for Research on Cancer in Group 1 of human carcinogens [20,21]. This mycotoxin is known to be produced by two mould species of the *Aspergillus* genus, i.e., *Aspergillus flavus* and *Aspergillus parasiticus*. OTA falls within the 2B Group of possible human carcinogens [22], and is produced by various mould species of both the *Aspergillus* and the *Penicillium* genus [16,23]. It has been established that OTA poses as the major meat contaminant, while $AFB_1$ is detected less often and in lower concentrations [2]. According to the European databases on undesirable residues in food, OTA is regularly found in the kidneys of slaughtered animals [24]. Pork consumption has historically been a significant source of human exposure to OTA, primarily in the East Europe. Unlike other mycotoxins, OTA has the potential to bioaccumulate in a monogastric organism, which explains its common presence in edible pig tissues and pork meat [25,26].

Numerous studies have demonstrated that certain temperatures, pH-values, water activity, casing cracking, the presence or absence of crust (in case of prosciuttos) or cracks and insufficient washing and brushing of dry-cured meat product surfaces (i.e., an uncontrolled mould growth), encourage superficial moulds of the *Penicillium* and the *Aspergillus* genera to produce the mycotoxins referred to above [7,23,27,28]. This pathway of meat product contamination is deemed significant.

On top of $AFB_1$ and OTA, some mould species are capable of producing other mycotoxins, such as citrinin (CIT), cyclopiazonic acid (CPA) and sterigmatocystin (STC), but their impact on the quality and safety of meat products and ultimately human health has remained unclarified [29]. STC represents a carcinogenic $AFB_1$ precursor that shares the bio-synthesis pathway with aflatoxins, and is produced by *Aspergillus* moulds, most commonly *Aspergillus versicolor* [30]. CPA can cause severe gastrointestinal and neurological disorders [31], and is synthesised by numerous *Penicillium* and *Aspergillus* moulds, in particular *Penicillium commune* [32–34]. The latter was isolated from the surfaces of various meat products, including European dry-fermented sausages and prosciuttos [35–37].

Research on mycotoxins in meat products, conducted in European countries, has mostly been focused on OTA and $AFB_1$, while the prevalence of, and consumer exposure to CIT, STC and CPA are still under-investigated, although their toxicity is unquestionable [29,30,38]. Some authors warn about a significant CPA presence in meat products [29,39,40],

so a sensitive and specific analytical technique was developed, confirming its high presence in different dry-cured meat products [41]. It is beyond doubt that a traditional meat product produced in a household should reveal not only mycotoxin contamination, but also the mycotoxin-producing mould species and factors of relevance for mycotoxin production, such as regional climate and production & storage conditions [42,43].

## 3. Mycotoxin Contamination through Spices

Little et al. [44] pointed out that the increase in processed food production and high meat demand are the major reasons behind the rapid increase in spices' consumption. So many traditional (Figure 1) and non-traditional spices are used in meat products' production to provide for distinctive flavours. Among the many, red, white, and black pepper, sweet and hot pepper, mustard, garlic, allspice, laurel, rosemary, and cinnamon are the most commonly used worldwide.

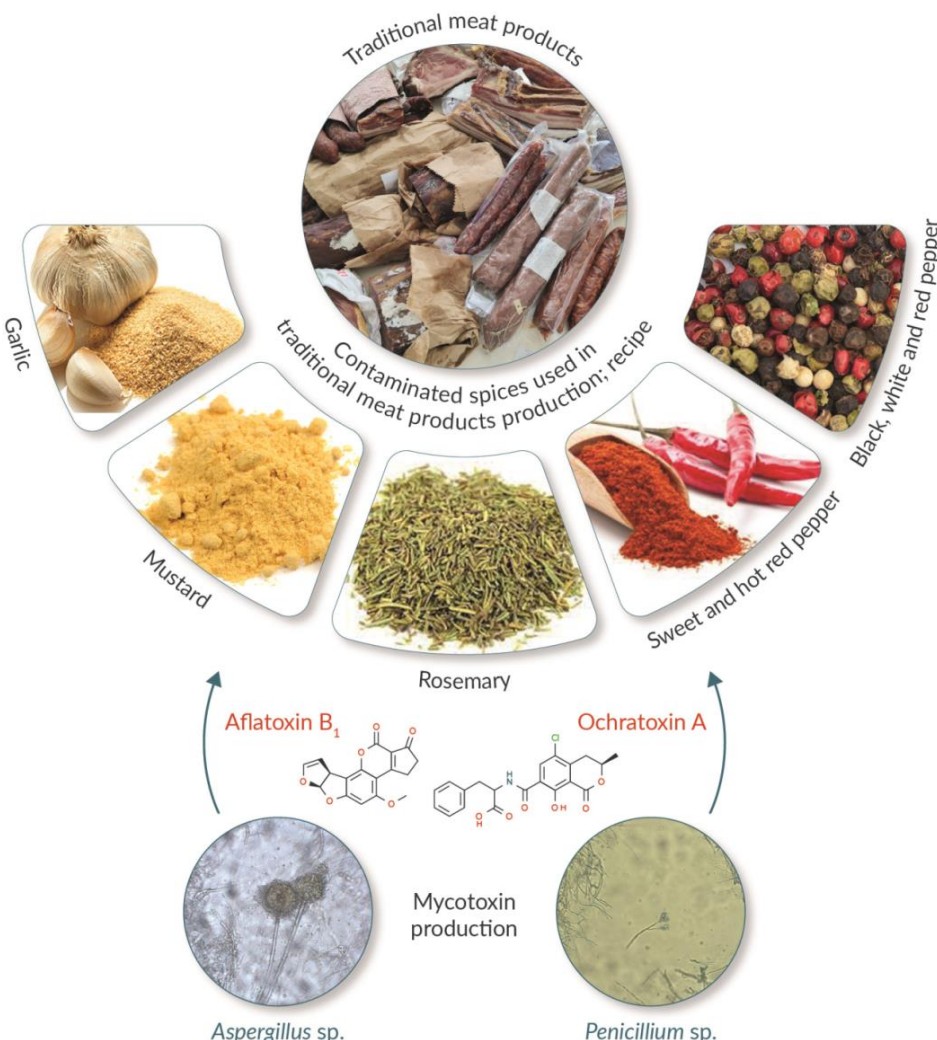

**Figure 1.** The use of different spices in meat production.

Spices are mainly imported from developing countries with tropical and subtropical climate, where high temperature, heavy rainfall and humidity often promote fungal growth and facilitate mycotoxin occurrence [45]. Some spices are especially susceptible to toxigenic moulds and mycotoxin development [46]. The most frequent fungal contaminants of spices are *Aspergillus* and the *Penicillium* species [47–51]. For instance, red pepper may contain *Aspergillus* mould spores, which then overgrow on fermented sausages. Janković et al. [52] analysed 15 different spices for the presence of xerophilic moulds and found that black and white grounded pepper were the most contaminated.

Furthermore, spices are usually left to dry on the ground out in the open. Poor outdoor hygiene further promotes mould growth and mycotoxin production [53]. Pickova et al. [45] demonstrated that chili, nutmeg, and paprika powders are the most problematic, since they often contain AFs, OTA and other mycotoxins in the amounts exceeding the maximum EU limits. However, as compared to AFs and OTA, other mycotoxins possibly present in spices have been insufficiently studied, so that their share in the supply of mycotoxins is difficult to evaluate. Spices purchased in open markets are usually significantly more contaminated than those coming from supermarkets [54].

High AFB$_1$ concentrations have been reported in paprika (155.7 μg/kg) [55] and black pepper (75.8 μg/kg) [56]. This applies to OTA, as well (177.4 μg/kg in paprika) [55]; 79.0 μg/kg in black pepper [57]. Pleadin et al. [58] found OTA levels of up to 8.11 μg/kg in red paprika used to spice dry-fermented sausages. A higher prevalence of OTA in prosciutto samples was sometimes linked to pepper spiking; namely, pepper often becomes contaminated with *Aspergillus* moulds, out of which *A. niger* produces OTA [19,59]. However, some studies have revealed that spices may also inhibit mould growth [60], resulting, for instance, in lower OTA contamination of some meat products [19].

## 4. Transfer of Mycotoxins by Carry-Over Effect

Natural mycotoxin contamination of cereals and feed was frequently demonstrated worldwide [61]. Research has shown that the presence of mycotoxins in food of animal origin may be consequential to farm animal feeding on contaminated feed (the carry-over effect) [9,10]. The effect is defined as the passage of undesired compounds from contaminated feed into food of animal origin (Figure 2), while the research topics of interest include the background and the mechanisms of this process, carry-over ratios and the resulting human health risks. With regard to the transfer into products of animal origin, only the basics of mycotoxin activity have been elucidated. In many cases, standardised parameters for the calculation of carry-over still so not exist, and trials are usually incomparable due to the different study designs [62].

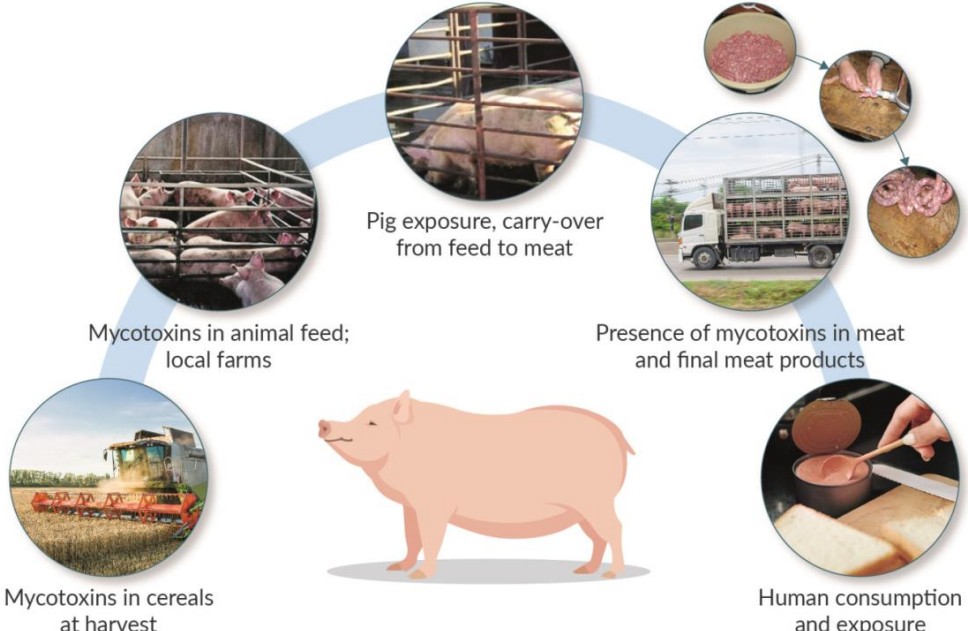

**Figure 2.** Carry-over of mycotoxins from contaminated cereals and feed into final meat products.

Transfer ratios vary; when it comes to skeletal muscles, the values are below 1%, while in the case of blood serum and fat tissue they are higher. Due to their detoxification role, higher carry-over ratios can be seen in liver and kidney. Once taken up by the host organism, these toxins initially reach the blood stream, where they can be found in measurable concentrations. Völkel et al. [62] concluded that carry-over ratios do not vary

only across toxin types and animal host species, but also across various tissues sampled from a single host.

It is known that an animal organism fed on contaminated feedstuffs or raw materials sets enzymatic and microbial transformations in motion, leading to the formation of gut metabolites. These metabolites can be absorbed into the bloodstream and later excreted through urine and feces, but their residues can be left behind in edible organs and muscles [63]. However, mycotoxin transfer through the carry-over effect still remains to be clarified by future research. In general, the residues of mycotoxins such as zearalenone, trichothecenes and fumonisins are not considered to be of public health importance, since the levels of these toxins found in tissues of animals experimentally fed on very high doses of the latter were low [24].

OTA has been described as a common mycotoxin encountered in animal-derived products. It is known that this mycotoxin can accumulate in animal meat and organs because of its high bioavailability and long half-life in some monogastric farm animals. Its transfer to beef has been a controversial subject, but studies generally show that bacterial metabolism in the gastrointestinal tract, especially the rumen, yields the less toxic cleavage product ochratoxin $\alpha$. No measurable carry-over into beef was detected [64], but this might only be true for full-grown animals with a well-developed digestive system. For calves, it was evidenced that carry-over ratios of mycotoxins are similar to those seen in monogastric organisms [62]. Poultry is generally less susceptible to OTA, since birds excrete OTA faster than mammals, thus lowering the chance for its accumulation. The half-life of OTA in broiler chickens is significantly shorter than in pigs (4.1 h versus 150 h), leading to a lower systemic exposure [65]. Milicevic et al. [66] concluded that, in comparison to other sources, chicken products-based OTA intake makes a negligible contribution to the total consumer OTA intake. However, a significant transfer of this mycotoxin from contaminated feed into raw biological materials was detected in pigs, recognized as the species most sensitive to OTA. It was therefore recommended to study OTA occurrence in pig blood and tissues, so as to assess the significance for animal and human health [16,67].

Pfohl-Leszkowicz and Manderville [68] pointed out that OTA concentrations in various tissues primarily depend on the length of exposure, the dosage, and the entry route. OTA has been detected in meat and its by-products, particularly in the kidney and whole blood/plasma [9,10,64,69]. This might be of special concern for the processing of local traditional meat products, such as blood pudding, lunchmeat, and sausages, since pig blood or plasma are added to many of them [62].

According to the 2006 EFSA study, the mean OTA contamination level in edible porcine offal ranged from 0.17 to 0.20 µg/kg [70]. Rosi et al. [69], who had administered 200 µg OTA/kg of feed, determined an OTA concentration of 9.6 ± 2.7 µg/kg in the kidney, 6.3 ± 1.7 µg/kg in the liver and 1.9 ± 0.6 µg/kg in the muscle tissue, while the lowest concentration of 1.1 ± 0.6 µg/kg was found in the adipose tissue. After a 40-day OTA treatment, a quite similar OTA concentration (2.21 ± 0.78 µg/kg) was determined in the pig muscle tissue by [71]. Ninety-day pig treatment with 100 µg OTA/kg of feed resulted in an OTA concentration of 7.9 µg/kg in the liver, 2.7 µg/kg in muscles and 16.2 µg/kg in the lungs [72]. A 40-day pig feeding on 0.68 mg OTA per day resulted in smoked ham OTA concentrations of 1.255 to 5.645 µg/kg, explained by the carry-over effect [73]. Perši et al. [10] evaluated OTA carry-over in raw materials and cooked meat products after pig sub-chronic exposure to 300 µg OTA/kg feed for 30 days. The results showed the highest OTA concentrations in the kidney, the lungs, the liver, the blood, the spleen, the heart, and the adipose tissue. As for cooked meat products, the highest average OTA concentrations were detected in black pudding sausages (14.02 ± 2.75 µg/kg), liver sausages (13.77 ± 3.92 µg/kg), and pâté (9.33 ± 2.66 µg/kg).

Evidence of carry-over of aflatoxins has so far been found in milk, porcine tissue, and eggs. Several cases of carry-over in swine have been reported for aflatoxin $B_1$. It may occur in liver, muscles, and adipose tissue. The addition of chemicals, such as aluminosilicate sorbents, to the fodder can decrease the amount of aflatoxin $M_1$ detectable in liver, kidney,

and muscle tissue, whereas the amount of aflatoxin $B_1$ only decreases in muscle tissue, but not in liver or kidney [74].

Little is known about the importance of carry-over of mycotoxins like citrinin, whose major impact on human health was believed to be the consumption of directly contaminated plant-derived products. The study of Meerpoel et al. [75] demonstrates that the presence of CIT in animal feed can lead to its transfer into edible pig tissues, although with low carry-over rates (0.1 and 2%) and that this toxin tends to accumulate in pig tissues. It was demonstrated that pig exposure to 1 mg CIT/kg feed for three weeks does not cause toxic effects in the kidney, liver, jejunum, or duodenum. However, mitochondrial changes that could be related to oxidative stress, which is the key mechanism of CIT toxicity, were detected.

## 5. Mycotoxin Production by Surface Moulds

Mycotoxin-producing moulds tend to grow on substrates of both plant and animal origin and prefer to inhabit regions characterised by constantly high relative air humidity and moderate to high temperatures. Mould genera most commonly linked to the nascence of mycotoxins are *Aspergillus, Penicillium, Fusarium, Claviceps, Alternaria, Pithomyces, Phoma, Stachybotrys* and *Diploidia*, out of which *Aspergillus, Penicillium* and *Fusarium* are the most represented [1,61]. Mould species of the outermost relevance for the production of mycotoxins on dry-cured meat product surfaces and conditions facilitating the production are shown in Table 1.

**Table 1.** Conditions under which certain moulds produce mycotoxins on the surface of dry-cured meat products [76,77].

| Mould Species | Mycotoxin | $a_w$ (Range) | T/$°$C (Range) |
|---|---|---|---|
| *Aspergillus flavus* | AFB$_1$ | $\geq$0.84; $\geq$0.80 | 12–35 |
| *Aspergillus parasiticus* | AFB$_1$ | $\geq$0.84 | 12–35 |
| *Aspergillus ochraceus* | OTA | $\geq$0.87 | 12–35 |
| *Penicillium verrucosum* | OTA | $\geq$0.85 | 2–34 |
| *Penicillium nordicum* | OTA | - | 15–30 |
| *Penicillium commune* | CPA | $\geq$0.90 | 12–30 |

AFB$_1$-aflatoxin $B_1$; OTA-ochratoxin A; CPA-cyclopiazonic acid.

During ripening, the surface of dry-cured meat products becomes overgrown with moulds whose spores mostly come from ripening chambers. The overgrowth is intensified by ripening longevity and in traditional household environments lacking microbiological filters and pneumatic barriers, hence making temperature and relative air humidity uncontrollable. Conditions favouring mould growth on meat product surfaces are a temperature of 10–45 $°$C, pH of 1.5–10, aw of min 0.6, aerobic redox potential, and salt content of up to 20% [60].

The impact of superficial moulds on product quality may be beneficial in terms of moisture retaining and drying and incrustation prevention that may occur due to protein coagulation. Mould enzymes take part in fermentation and ripening, either alone or in combination with endogenous enzymes present in stuffing, but can also contaminate the products due to mycotoxin production [18,23,78–80]. Research has established that during long ripening of dry-fermented sausages (up to six months) and prosciuttos (12–18 months), mycotoxins may be produced by unwanted superficial mould species, which overgrow the ripening product [12,19].

It was documented that superficial moulds that spontaneously overgrow on product surfaces mostly belong to the *Penicillium* and the *Aspergillus* genera (Figure 3), the first thereby being encountered more frequently, as concluded by numerous European authors [78,81–84], and in wider variety [19]. The main mycotoxin producers are *Aspergillus ochraceus, Aspergillus versicolor, Penicillium nordicum* and *Penicillium verrucosum* [23,27,79,84–86], which are commonly found in salami and dry-cured hams [19,79,87].

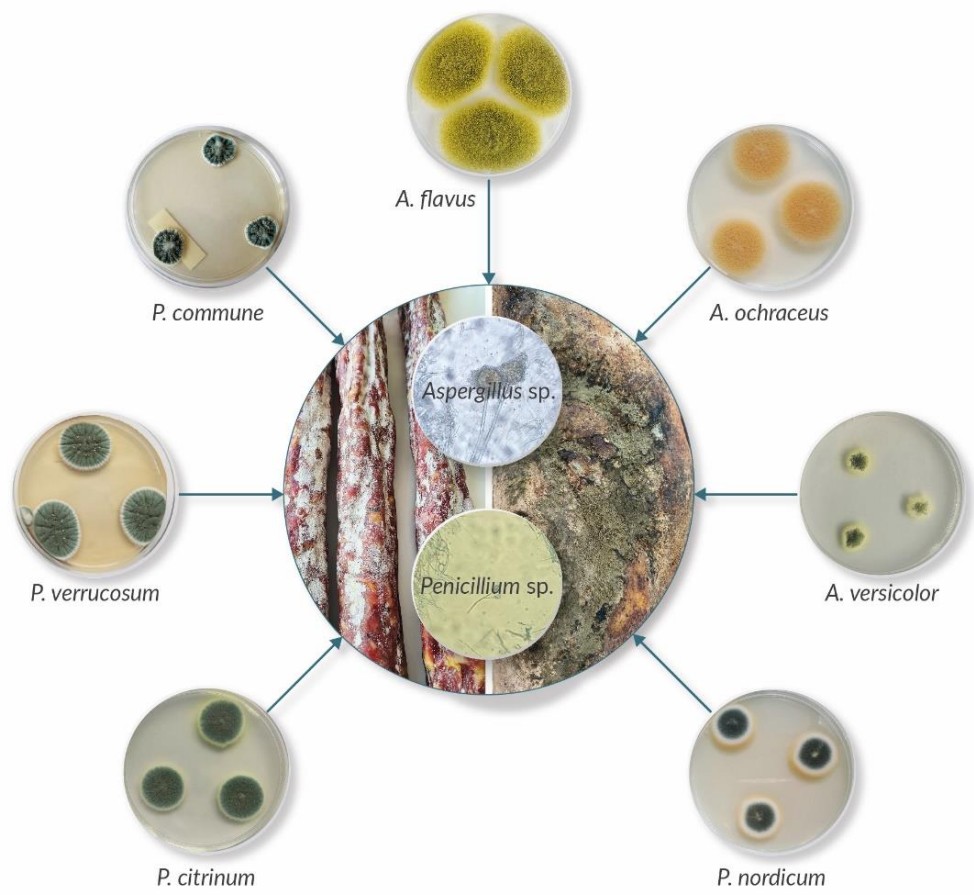

**Figure 3.** Superficial moulds of the *Penicillium* and the *Aspergillus* genera that overgrow on the surface of dry-cured meat products.

OTA is mainly produced by the *Aspergillus ochraceus*, while CIT mostly comes from the *Penicillium citrinum. Penicillium verrucosum* is capable of producing both of the above [88], so that on occasions the two mycotoxins may simultaneously be present in various food-stuffs [38,89]. *Penicillium nordicum* is a contaminant of protein-rich foods [72,84,86]. This mould can grow at low temperatures (15 °C), especially should the salt content be >5%, so that it is often isolated from refrigerated protein foods such as dry-cured ham, salami, and salted fish. It is not uncommon for mycotoxins to be present in meat products in substantial concentrations, as the cracking of the outer casing facilitates their diffusion into the interior of the product [11,17,72].

Matrella et al. [90] found 40% OTA positive dry-cured hams in which the mean OTA concentration of 4.06 μg/kg and the maximum concentration 28.4 μg/kg were found, and concluded that only direct contamination with OTA-producing strains during ripening can be behind such contamination. Pietri et al. [91] analysed north Italian pork products and found high OTA levels in 5 (17%) dry-cured hams in which OTA concentration exceeded 1.0 μg/kg. The authors also concluded that only direct OTA contamination can explain such high contamination levels, and blamed *P. verrucosum*, which was earlier proven to be present in industrial ripening chambers' atmosphere and on ham surfaces [81].

OTA was determined on surfaces of 84 out of 110 samples of different Italian hams in the concentration of 0.53 μg/kg, while the concentration in the innermost layers of 32 ham samples was lower than 0.1 μg/kg [72]. Iacumin et al. [23] investigated the occurrence of OTA in north Italian dry sausages and found them to be vastly contaminated from the outside (3–18 μg/kg), but not from the inside, indicating that intact casings shield against environmental contamination. Roncada et al. [92] found OTA on the surface (casing) of 25.0% of the sampled Italian artisan salamis, in two cases in concentrations exceeding the 1 μg/kg guidance value. The authors emphasised the importance of OTA presence on the

casings given that the products are often sliced together with the casing. The presence of mycotoxins on the surface of these products was linked to the presence of surface moulds.

## 6. Occurrence of Mycotoxins in Meat Products

Research on mycotoxins in meat products conducted thusfar has mostly been focused on OTA and $AFB_1$. Although the research we have conducted demonstrated meat product contamination with $AFB_1$ to be negligible, it is important to keep in mind that the research done in the last decade has documented an extremely high contamination of maize and feed mixtures with this mycotoxin [93–96], which might have contributed to the significant contamination of meat products. The above studies reported that $AFB_1$ and OTA contamination is to be attributed to the inadequate production control and storage practices, imposing the need for prevention, systematic control, and further monitoring.

To this point, no maximum levels (ML) applicable to mycotoxins in meat, offal, and meat products have been stipulated by regulatory bodies. The possibility of reducing mycotoxin levels using suitable household-practiced physical methods (cooking, baking) [94] and/or gamma irradiation [97] has been investigated, too. The research demonstrated these methods to be of limited value (20 to 30% efficiency) when it comes to OTA reduction, proving the need for the prevention of contamination rather than its reduction.

Markov et al. [98] investigated the presence of CIT in meat products and demonstrated its poor representation (only 4.44% positives) in concentrations close to the limit of detection of the analytical technique used (1.0–1.3 μg/kg). However, since CIT presence is climate-dependent, this mycotoxin deserves further and broader research. On top of that, the co-occurrence of CIT and OTA reported in the literature should be of concern as well.

The occurrence of mycotoxins evidenced in different types of meat products worldwide is shown in Table 2.

**Table 2.** The worldwide occurrence of mycotoxins in meat products.

| Product | Mycotoxin | N | % Positive Samples | Range (μgkg$^{-1}$) | Country | Reference |
|---|---|---|---|---|---|---|
| Beef luncheon, burger and sausages | AFs | 150 | 0.6 | 2–7 | Egypt | [99] |
| | | 50 | 14 | 11.1 | Egypt | [100] |
| | | 10 | 0 | <LOD | Spain | [101] |
| | | 25 | 100 | 0.47–9 | Egypt | [102] |
| Blood sausages | OTA | 620 | 77 | 3.2 | German | [13] |
| Liver-type sausage | | 620 | 68 | 5 | German | [13] |
| Sausages | | 100 | 45 | 7–8 | Italy | [23] |
| | | 10 | 0 | <LOD/LOQ | Spain | [101] |
| Parma (retail product) | | 22 | 4 | 56–158 | Denmark | [103] |
| Dry-cured Iberian ham | | 20 | Deep portion 15 | Deep portion 2–160 | Spain | [27] |
| | | 20 | Superficial portion 25 | Superficial portion > 15 | Spain | [27] |
| | | 45 | 13 | 1.9–6.3 | Spain | [27] |
| Beef luncheon | | 25 | 100 | 0.56–8.5 | Egypt | [102] |
| Beef burger | | 25 | 100 | 2.7–7.6 | Egypt | [102] |
| Fermented meat products | OTA | 90 | 64.44 | 1.23–7.83 | Croatia | [98] |
| | CIT | | 4.44 | 1.0–1.3 | | |
| | $AFB_1$ | | 10 | 1.0–3.0 | | |
| Traditional meat products | $AFB_1$ | 410 | up to 11.1 | up to 1.69 | Croatia | [17] |
| | OTA | | up to 20.0 | up to 9.95 | | |
| Traditional meat products | $AFB_1$ | 160 | 8 | up to 1.92 | Croatia | [19] |
| | OTA | | 14 | up to 6.86 | | |
| Dry-fermented sausages | OTA | 88 | 14.8 | up to 0.48 | Croatia | [104] |
| Dry-fermented sausages | CPA | 47 | 14.9 | 2.55–59.80 | Croatia | [41] |

N-number of samples analyzed; $AFB_1$-aflatoxin $B_1$; AFs-aflatoxins; OTA-ochratoxin A; CIT-citrinin; CPA-cyclopiazonic acid.

## 7. Control and Prevention

It is known that the presence of mycotoxins in food and feed has a negative economic and social impact in terms of human and animal deaths and diseases, veterinary and medical costs, reduced productivity, loss of livelihoods, food and feed loses and contamination [105]. The EU respected MLs of aflatoxins in food, in particular the level of $AFB_1$ and the total aflatoxin levels, are stipulated by the Commission regulations [106,107], but do not apply to meat, offal, and meat products. The ML for OTA in food is also defined by [106], but also does not apply to these foodstuffs. Furthermore, CIT, STC and CPA MLs in meat, offal, and meat products haven't been established yet. Denmark (10 µg/kg in pig kidney), Estonia (10 µg/kg in pig liver), Romania (5 µg/kg in pig kidney, liver, and meat), and Slovakia (5 µg/kg in meat) are exceptions, since OTA MLs are given under their national regulations, while Italy, for example, has adopted guideline OTA values (of 1 µg/kg in pig meat and products) [108].

Many countries lack national legislation governing this area, but producers should be aware of the possibility of the contamination of meat products and carry out systematic controls of mycotoxins via HACCP (Hazard Analysis and Critical Control Point) programs. By sending samples to authorized laboratories, which should use sensitive analytical methods validated for such matrices, they will enable the detection of possible mycotoxin (primarily $AFB_1$ and OTA) presence in dry-cured meat products, which are contaminated most often.

It is known that meat products' production technologies, such as heat treatment, salting, drying, and ripening, as well as storage, do not significantly reduce the amount of these toxins in final products [94,109,110]. Therefore, to prevent the occurrence of mycotoxins in meat products, the key factor is to control and prevent the use of contaminated raw materials. Mould growth has to be effectively controlled, bearing in mind that their activity is crucial for the sensory properties of these products [35,111,112]. Since $a_w$ of the substrate affects the ability of mould-delivered mycotoxin production, the latter can be prevented by controlling this parameter and adjusting the drying and ripening temperature. For example, significantly higher mycotoxin production was found with $a_w$ of 0.99 as compared to that of 0.97 or 0.95 [7].

Moulds should be continuously removed from the product surface during ripening either by brushing or by washing in order to prevent excessive mouldiness [82]. Manufacturers typically wash dry-cured meat products between drying and ripening to remove visible mould colonies [7], because consumers find mouldy products less appealing. It was found that the concentration of OTA on the outer surface (casing) of dry-fermented sausages treated by brushing and washing was lower than the limit of detection of the applied analytical method. It is also common to spray rice flour on the surface of ripe sausages after removing the mould layer by brushing, washing or under air pressure. Iacumin et al. [23] suggested that the surface of dry-fermented sausages should be first brushed and then washed, so as to lower OTA concentration and eliminate the potential consumer health threat.

In order to prevent excessive mouldiness of the product surface during ripening, it is necessary to ensure an adequate distance between the products so as to allow for an unhindered air flow. Maturation should take place in ripening chambers equipped with biological microfilters allowing for fresh air supply. Before the intake and ripening, ripening chamber surfaces should be coated with fungicidal coatings, while the entrance should be provided with a pressure barrier [113]. Uncontrolled meat product' production can mostly be seen in rural households, allowing for the significant influence of external factors that encourage the occurrence of toxicogenic surface moulds and hence the contamination of meat products.

Research conducted in European countries on certain types of dry-cured meat products [12,19,36,79,83,84] has tackled the issue of superficial mould presence and mycotoxin contamination only partly. In addition, physicochemical parameters of relevance for mould nascence, such as water activity, pH-value, and water and salt content [7,114,115] haven't

been investigated in full. The same goes for climate-conditioned mould prevalence in certain traditional meat products-producing geographical regions. Therefore, all these issues of relevance should be the subject of further research.

### 8. Conclusions

Research on mycotoxins in meat products conducted thusfar has mostly been focused on OTA and AFB$_1$, while the knowledge on CIT, STC and CPA prevalence is still insufficient. Research on human exposure to mycotoxins through meat products was targeted mainly to OTA, while the potential (and even concurrent) presence of other mycotoxins has not been investigated in depth. In order to establish meat products' mycotoxin contamination more precisely, the concentrations of all mycotoxins of relevance for meat products should be determined. To accomplish that goal, analytical techniques suitable for mycotoxins (especially STC and CPA, which have been under-investigated) determination should be developed. Once these analytical tools are developed, they shall make a major contribution to further research on the prevalence of mycotoxins in meat products and consumer exposure-related issues. Moulds isolated from the surfaces of various dry-cured meat products should be precisely identified using molecular techniques; in due course, their presence on the surface of meat products produced in households should be linked to environmental (climate) conditions. Based on the concentrations of all mycotoxins of importance for meat products, and based on the target population's dietary habits, future studies should aim at delivering a well-founded assessment of consumer exposure to mycotoxins consequential to meat product consumption. The results of such studies should also serve as a valuable rationale for establishing mycotoxin MLs in meat products and guidance for meat product producers that shall help them prevent the mycotoxin contamination of their products.

**Author Contributions:** Conceptualization, J.P.; resources, D.M., B.Š. and I.K.; writing—original draft preparation, J.P., T.L. and D.M.; writing—review and editing, K.M. and N.V.; visualization, T.L.; supervision, K.M. and M.Z.; project administration, J.P. All authors have read and agreed to the published version of the manuscript.

**Funding:** This research was funded by the Croatian Science Foundation to the project "Mycotoxins in traditional Croatian meat products: molecular identification of mycotoxin-producing moulds and consumer exposure assessment" (No. IP-2018-01-9017).

**Institutional Review Board Statement:** Not applicable.

**Informed Consent Statement:** Not applicable.

**Conflicts of Interest:** The authors declare no conflict of interest.

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
