# Peer review of "Pathways of Mycotoxin Occurrence in Meat Products: A Review"

_processes, doi:10.3390/pr9122122_

Round 1

Reviewer 1 Report

Dear Authors,

This paper is a well-written review with a large number of references. There is no doubt that you have put a lot of effort into writing the paper. However, I cannot see a clear connection between the title of the paper and its content. Much of the paper is about meat and all meat products, not just traditional products. I suggest you remove the word "traditional" from the title and restructure the paper by writing about meat products and TMP.
Other comments:
- Line 134; the parentheses are missing
- Line 138; a comma is missing
- Regarding the pictures: Are they of yourself?

Author Response

Comment: This paper is a well-written review with a large number of references. There is no doubt that you have put a lot of effort into writing the paper. However, I cannot see a clear connection between the title of the paper and its content. Much of the paper is about meat and all meat products, not just traditional products. I suggest you remove the word "traditional" from the title and restructure the paper by writing about meat products and TMP.

Answer:  In line with the Referee’s comment, the manuscript title has been changed; changes in this regard have been made throughout the body text, as well.

Comment:
Line 134; the parentheses are missing.

Answer:  The missing parenthesis has been added.

Comment:  
Line 138; a comma is missing.

Answer:  The missing comma has been added.

Comment:  Regarding the pictures: Are they of yourself?

Answer:  All three Figures are self-authored and compiled to the effect of as clear presentation of meat products’ contamination pathways as possible. The Figures were compiled for the sole purpose of graphic supplementation of this manuscript and illustration of its key points.

Reviewer 2 Report

This MS basically summarize the occurrence of mycotoxins in meat and meat products; accordingly, I believe the authors should change the title. First of all, this MS must be subject to extensive language editing, formatting and reforming. I would like to draw many mistakes and not-accurate data in just one paragraph:

“line 30-31: coming as secondary metabolites”, “line 33: No matter how undesirable”, “Although a low-level exposure may not be a major health concern” are you sure that low level of mycotoxins is not toxic?!! Aflatoxin B1 is the most potent natural toxins that classified as group one carcinogen and very low level is very dangerous, food with amount higher than 2-20 ppb is not fit for human consumption. Line 37, so foth,??? Line 42 “such as the mycotoxin in question and analytical or reporting methods used”, what is mycotoxins in question??

Combine figure one and two, direct and indirect method for mycotoxin contamination in meat and meat products, revise Fig 3, how about A. flavus? Source of pictures? Can you include the growth of these fungi on meat samples not media?

Its is nice to list the spices that could contaminated by mycotoxins and regularly used in meat, but I would clarify more and focus on the contamination of mycotoxins due to feeding of animals of food-contaminated with mycotoxins.

Can you clarify what you meant by this Table 1 “Table 1.Conditions under which certain moulds PRODUCE MEAT PRODUCTS-CONTAMINATING MYCOTOXINS”???

Table 2, include source of contamination.

The subject is not novel but appealing, however, I believe this MS needs extensive revision and re-work. More related research articles, not review paper, is needed.

Author Response

Comment:  This MS basically summarize the occurrence of mycotoxins in meat and meat products; accordingly, I believe the authors should change the title.

Answer:  In line with the opinion of both of our Referees, the manuscript title has been changed. This review article aimed at presenting the knowledge on potential sources and pathways of mycotoxin contamination of various meat products.

Comment:  First of all, this MS must be subject to extensive language editing, formatting and reforming. I would like to draw many mistakes and not-accurate data in just one paragraph:

“line 30-31: coming as secondary metabolites”, “line 33: No matter how undesirable”, “Although a low-level exposure may not be a major health concern” are you sure that low level of mycotoxins is not toxic?!! Aflatoxin B1 is the most potent natural toxins that classified as group one carcinogen and very low level is very dangerous, food with amount higher than 2-20 ppb is not fit for human consumption. Line 37, so foth,??? Line 42 “such as the mycotoxin in question and analytical or reporting methods used”, what is mycotoxins in question??

Answer:  The manuscript has been revised line-by-line according to the above comments. The sentences have mostly been altered by virtue of deletion of certain parts (words). The revised manuscript has been rechecked by a highly qualified linguist (University Professor) having a vast experience in scientific writing and MS reviewing. The revised manuscript strictly follows the instructions for authors submitting their contributions to the “Processes”. Please accept our apologies for the incompliance of the original manuscript with the journal guidelines; we suspect that the merging of a large number of words that inadvertently occurred in the original submission may have come as a result of incompatibility of PCs used by authors during their simultaneous work on the MS.

Comment:  Combine figure one and two, direct and indirect method for mycotoxin contamination in meat and meat products.

Answer:  The authors are of the opinion that, should Figures 1 and 2 be merged, the merged figure would be too large and unclear, given that each of the original figures contains a number of smaller ones. Such a merging would also substantially alter the MS concept. Whereas the MS was originally divided into subtitles in order to present all three possible meat products’ contamination pathways, the figures in question were meant to separately illustrate the key points under the subtitles “Mycotoxin contamination through spices” and “Transfer of mycotoxins by carry-over effect”. Therefore, we would be most obliged to our esteemed Referee if he/she would reconsider the suggestion and accept our explanation.

Comment:  Revise Fig 3, how about A. flavus?

Answer:  Figure 3 has been revised in line with the Referee’s comment. A. flavus, as a significant potential source of contamination of dry-cured meat products, has been added on the top of the Figure.  

Comment:  Source of pictures?

Answer: All three Figures are self-authored and compiled to the effect of as clear presentation of meat products’ contamination pathways as possible. The Figures were compiled for the sole purpose of graphic supplementation of this manuscript and illustration of its key points.

Comment:  Can you include the growth of these fungi on meat samples not media?

Answer:  In view of the Referee’ s comment, we would like to respond by offering an explanation. Based on our experience gathered insofar and based on the literature data, mould species overgrowing the surface of dry-cured meat products are impossible to be accurately pinpointed due to the manner of their growth (i.e., the tendency to grow one over another, as well as due to their unspecific morphology). Therefore, to accurately identify mould species overgrowing a given product surface, pure cultures should be isolated. For this reason, our review article elaborates only pure cultures of isolated and identified mycotoxigenic moulds (isolated on meat products within the frame of own research).     

Comment:  Its is nice to list the spices that could contaminated by mycotoxins and regularly used in meat, but I would clarify more and focus on the contamination of mycotoxins due to feeding of animals of food-contaminated with mycotoxins.

Answer:  We are most obliged to our Referee for this comment. We have put additional effort to further clarify mycotoxin contamination arising on the grounds of animal feeding on food contaminated with mycotoxins under the subtitle “Transfer of mycotoxins by carry-over effect“. The sentences added to the original text and two additional citations have been made visible using the Track Changes mode.

Comment:  Can you clarify what you meant by this Table 1 “Table 1.Conditions under which certain moulds PRODUCE MEAT PRODUCTS-CONTAMINATING MYCOTOXINS”???

Answer:  The title of the Table 1 has been changed to read as follows: “Conditions under which certain moulds produce mycotoxins on the surface of dry-cured meat products”.

Comment:  Table 2, include source of contamination.

Answer:  All literature sources included into this table have been rechecked; it came to our attention that the majority of the research studies failed to clearly identify the source of mycotoxin contamination of meat products. Namely, the authors of the studies in question mostly reported on the prevalence of mycotoxins rather than the contamination source, which remained either unsearched for or unknown. In view of the foregoing, a column in the Table 2 containing data on contamination source would be virtually empty and was therefore omitted. We fully agree with our esteemed Referee that data on contamination sources would be more than useful and would add value to this review article, but regrettably most of the studies carried out insofar do not offer these data.      

Comment:  The subject is not novel but appealing, however, I believe this MS needs extensive revision and re-work. More related research articles, not review paper, is needed.

Answer:  This paper aims at bringing the overview of the recent literature dealing with mycotoxin contamination of meat products through three possible contamination pathways. Supplemented with original, self-authored figures given under the subtitles, which aim at illustrating each of the three contamination pathways, we do believe that this review article is of value to the scientific community. We have made every effort to revise the manuscript in line with the suggestions and comments of both Referees and to improve its technical and linguistic aspect. Our research on direct mould contamination of meat products and its relationship with regional climate witnessed in the production areas is currently in progress, so that we plan to publish research works hopefully contributing to this field of science & research with some novel notions.  

The authors are indebted to our esteemed Reviewers for their most helpful suggestions and comments, to which we made every effort to fully respond. 

Reviewer 3 Report

Manuscript ID: processes-1414006
Type of manuscript: Review
Title: Pathways of mycotoxin occurrence into traditional meat products: a
review

Dear Authors,

The article is interesting, easy to follow and it contributes significantly to the meat processing area.

Line 35. Please remove the word “international”.

Line 55. I am not sure if the words “of note” are correct here. Do you agree to better delete them?

Line 106. Please change “expose to, CIT, STC and CPA ……” to “expose to CIT, STC and CPA”...

Line 126. Change the sentence “which then overgrow fermented sausages.” To “which then overgrow on fermented sausages.”

Line 184-185. Please change the sentence “For calves, it was evidenced that carry-over ratios are similar to those seen in monogastric organisms” to “For calves, it was evidenced that carry-over ratios of mycotoxins are similar to those seen in monogastric organisms”.

Line 186. Use the word “thus” instead of “hence” as I think it is more apporpriate.

Line 262=263. Please change “moulds that spontaneously overgrow product surfaces” to “moulds that spontaneously overgrow on product surfaces”

Line 284-285. In my opinion, there is not need to add the word “by” as the sentence is followed by the reference in brackets. “earlier proven to be present in industrial ripening chambers’ atmosphere and on ham surfaces by [81].

Line 299-230. Please change the figure title “Figure 3. Superficial moulds of the Penicillium and the Aspergillus genera that overgrow the surface of traditional dry-cured meat products” to “Figure 3. Superficial moulds of the Penicillium and the Aspergillus genera that overgrow on the surface of traditional dry-cured meat products”.

Line 318. Please change “Markov et al. [9698] investigated into the presence of CIT in meat products” to “Markov et al. [9698] investigated the presence of CIT in meat products”.

Line 369-370. Please uniform the writing in both, “Limit of Detection” and “Limit-of- Detection” (line 320).

Line 401-403. Please change the sentence “These newly developed analytical tools shall make a major contribution …. “ To “Once these analytical tools are developed, they shall make a major contribution….”

Author Response

Dear Editor,

Taking into consideration all sugestions given by the Reviewer, we have made the following amendments.

Comment: Line 35. Please remove the word “international”.

Answer: The word “international“ is now removed.

Comment: Line 55. I am not sure if the words “of note” are correct here. Do you agree to better delete them?

Answer: The term „of note“ is now deleted.

Comment: Line 106. Please change “expose to, CIT, STC and CPA ……” to “expose to CIT, STC and CPA”...

Answer: The comma in this sentence has been deleted.

Comment: Line 126. Change the sentence “which then overgrow fermented sausages.” To “which then overgrow on fermented sausages.”

Answer: The sentence was changed.

Comment: Line 184-185. Please change the sentence “For calves, it was evidenced that carry-over ratios are similar to those seen in monogastric organisms” to “For calves, it was evidenced that carry-over ratios of mycotoxins are similar to those seen in monogastric organisms”.

Answer: The sentence was changed.

Comment: Line 186. Use the word “thus” instead of “hence” as I think it is more apporpriate.

Answer: The word “thus” is used instead of “hence”.

Comment: Line 262-263. Please change “moulds that spontaneously overgrow product surfaces” to “moulds that spontaneously overgrow on product surfaces”.

Answer: The sentence was changed.

Comment: Line 284-285. In my opinion, there is not need to add the word “by” as the sentence is followed by the reference in brackets. “earlier proven to be present in industrial ripening chambers’ atmosphere and on ham surfaces by [81].

Answer: A suggested change has been made.

Comment: Line 299-230. Please change the figure title “Figure 3. Superficial moulds of the Penicillium and the Aspergillus genera that overgrow the surface of traditional dry-cured meat products” to “Figure 3. Superficial moulds of the Penicillium and the Aspergillus genera that overgrow on the surface of traditional dry-cured meat products”.

Answer: A suggested change has been made.

Comment: Line 318. Please change “Markov et al. [9698] investigated into the presence of CIT in meat products” to “Markov et al. [9698] investigated the presence of CIT in meat products”.

Answer: A suggested change has been made.

Comment: Line 369-370. Please uniform the writing in both, “Limit of Detection” and “Limit-of- Detection” (line 320).

Answer: A suggested change has been made.

Comment: Line 401-403. Please change the sentence “These newly developed analytical tools shall make a major contribution …. “ To “Once these analytical tools are developed, they shall make a major contribution….”

Answer: A suggested change has been made.

In the revised version, changes to the original manuscript have been made visible using the Track Changes option.

The authors are indebted to the Reviewer for his/her helpful suggestions and comments.

Looking forward to hearing from you,

Sincerely yours,

Manuela Zadravec

Round 2

Reviewer 1 Report

Thank you, you answer all my questions.

Author Response

The authors are indebted to our esteemed Reviewer for their most helpful suggestions and comments, to which we made every effort to fully respond. 

Reviewer 2 Report

Thank you for replying the reviewer comments and making some correction, however, the MS still need extensive English not only editing, but reforming and re-structure. Just as an example, In the abstract, "In order to establish  meat products’ mycotoxin contamination more precisely, the concentrations of all mycotoxins of relevance for these products should be determined". Can you get any point from this sentence? Pictures need to revise in a coherent and useful way. The authors must include more research articles and come up with clear way to present the data.

Author Response

Revision 2

Comments of the Referee 2:

Comment: Thank you for replying the reviewer comments and making some correction, however, the MS still need extensive English not only editing, but reforming and re-structure. Just as an example, In the abstract, "In order to establish  meat products’ mycotoxin contamination more precisely, the concentrations of all mycotoxins of relevance for these products should be determined". Can you get any point from this sentence? Pictures need to revise in a coherent and useful way. The authors must include more research articles and come up with clear way to present the data.

Answer: We believe that on the basis of the above paragraph as it is written, Editors can clearly understand that the review has been done completely unprofessionally, unclearly and without definition. As a result we cannot make any changes to our text in accordance with the above-mentioned comments. Therefore, we kindly ask you to send our manuscript to a third reviewer for their opinion.